# NOUGAT: NEURAL OPTICAL UNDERSTANDING FOR ACADEMIC DOCUMENTS

**Lukas Blecher**[*]  **Guillem Cucurull**  **Thomas Scialom**  **Robert Stojnic**

**Meta AI**

## ABSTRACT

Scientific knowledge is predominantly stored in books and scientific journals, often in the form of PDFs. However, the PDF format leads to a loss of semantic information, particularly for mathematical expressions. We propose Nougat (**N**eural **O**ptical **U**nderstanding for **A**cademic Documen**t**s), a Visual Transformer model that performs an *Optical Character Recognition* (OCR) task for processing scientific documents into a markup language, and demonstrate the effectiveness of our model on a new dataset of scientific documents. The proposed approach offers a promising solution to enhance the accessibility of scientific knowledge in the digital age, by bridging the gap between human-readable documents and machine-readable text. We release the models and code to accelerate future work on scientific text recognition.

## 1 INTRODUCTION

The majority of scientific knowledge is stored in books or published in scientific journals, most commonly in the Portable Document Format (PDF). Next to HTML, PDFs are the second most prominent data format on the internet, making up 2.4% of common crawl (Sebastian Spiegler, 2013). However, the information stored in these files is very difficult to extract into any other formats. This is especially true for highly specialized documents, such as scientific research papers, where the semantic information of mathematical expressions is lost.

Existing Optical Character Recognition (OCR) engines, such as Tesseract OCR (Smith, 2007), excel at detecting and classifying individual characters and words in an image, but fail to understand the relationship between them due to their line-by-line approach. This means that they treat superscripts and subscripts in the same way as the surrounding text, which is a significant drawback for mathematical expressions. In mathematical notations like fractions, exponents, and matrices, relative positions of characters are crucial.

Converting academic research papers into machine-readable text also enables accessibility and searchability of science as a whole. The information of millions of academic papers can not be fully accessed because they are locked behind an unreadable format. Existing corpora, such as the S2ORC dataset (Lo et al., 2020), capture the text of 12M[1] papers using GROBID (Lopez, 2023), but are missing meaningful representations of the mathematical equations.

To this end, we introduce Nougat, a transformer based model that can convert images of document pages to formatted markup text.

The primary contributions in this paper are

- Release of a pre-trained model capable of converting a PDF to a lightweight markup language. We release the code and the model on GitHub[2]

- We introduce a pipeline to create dataset for pairing PDFs to source code

- Our method is only dependent on the image of a page, allowing access to scanned papers and books

---

[*]Correspondence to: lblecher@meta.com

[1]The paper reports 8.1M papers but the authors recently updated the numbers on the GitHub page https://github.com/allenai/s2orc

[2]https://github.com/facebookresearch/nougat

## 2 RELATED WORK

Optical Character Recognition (OCR) is an extensively researched field in computer vision for a variety applications, such as document digitalization (Moysset et al., 2017; Smith, 2007), handwriting recognition and scene text recognition (Bautista & Atienza, 2022; Li et al., 2022; Diaz et al., 2021). More concretely, recognizing mathematical expressions is a heavily researched subtopic. Grammar based methods (MacLean & Labahn, 2013; Awal et al., 2014; Álvaro et al., 2014) for handwritten mathematical expressions were improved upon by different encoder-decoder models. The fully convolutional model (Yan et al., 2020) was succeeded by various RNN decoder models (Deng et al., 2016; Le & Nakagawa, 2017; Singh, 2018; Zhang et al., 2018; Wang & Liu, 2019), both for handwritten and printed formulas. Recently, the decoder (Zhao et al., 2021; Mahdavi et al., 2019) as well as the encoder (Blecher, 2023) were replaced with the Transformer (Vaswani et al., 2017) architecture.

Visual Document Understanding (VDU) is another related topic of deep learning research and focuses on extracting relevant information of a variety of document types. Previous works depend on pre-trained models that learn to extract information by jointly modeling text and layout information using the Transformer architecture. The LayoutLM model family (Xu et al., 2020; 2022; Huang et al., 2022) uses masked layout prediction task to capture the spatial relationships between different document elements.

Open source solutions with a related goal as ours include GROBID (Lopez, 2023), which parses digital-born scientific documents to XML with a focus on the bibliographic data and `pdf2htmlEX` (Lu Wang & Wanmin Liu, 2013), that converts digital-born PDFs to HTML while preserving the layout and appearance of the document. However, both solutions can not recover the semantic information of mathematical expressions.

Previous VDU methods either rely on OCR text from a third party tool (Xu et al., 2020; 2022; Appalaraju et al., 2021) or focus on document types such as receipts, invoices or form-like documents (Majumder et al., 2020). Recent studies (Kim et al., 2022; Davis et al., 2022) show that an external OCR engine is not necessarily needed to achieve competitive results in VDU.

## 3 MODEL

The architecture is a encoder-decoder transformer (Vaswani et al., 2017) architecture, that allows for an end-to-end training procedure. We build on the Donut (Kim et al., 2022) architecture. The model does not require any OCR related inputs or modules. The text is recognized implicitly by the network. See Fig. 1 for an overview of the approach.

**Encoder** The visual encoder receives a document image $\mathbf{x} \in \mathbb{R}^{3 \times H_0 \times W_0}$, crops the margins and resizes the image to fit in a fixed rectangle of size $(H, W)$. If the image is smaller than the rectangle, additional padding is added to ensure each image has the same dimensionality. We use a Swin Transformer (Liu et al., 2021), a hierarchical vision transformer (Dosovitskiy et al., 2021) that splits the image into non-overlapping windows of fixed size and applies a series of self-attention layers to aggregate information across these windows. The model output a sequence of the embedded patches $\mathbf{z} \in \mathbb{R}^{d \times N}$ where $d$ is the latent dimension and $N$ is the number of patches.

**Decoder** The encoded image $\mathbf{z}$ is decoded into a sequence of tokens using a transformer decoder architecture with cross-attention. The tokens are generated in an auto-regressive manner, using self-attention and cross-attention to attend to different parts of the input sequence and encoder output respectively. Finally, the output is projected to the size of the vocabulary $v$, yielding the logits $\ell \in \mathbb{R}^v$. Following Kim et al. (2022), we use the implementation of the mBART (Lewis et al., 2019) decoder. We use the same tokenizer as Taylor et al. (2022) because their model is also specialized in the scientific text domain.

### 3.1 SETUP

We render the document images at a resolution of 96 DPI. Due to the restrictive possible input dimensions of the Swin Transformer we choose the input size $(H, W) = (896, 672)$. The aspect ratio is in between the US letter and Din A4 format $\frac{22}{17} < \frac{4}{3} < \sqrt{2}$. The document images are resized

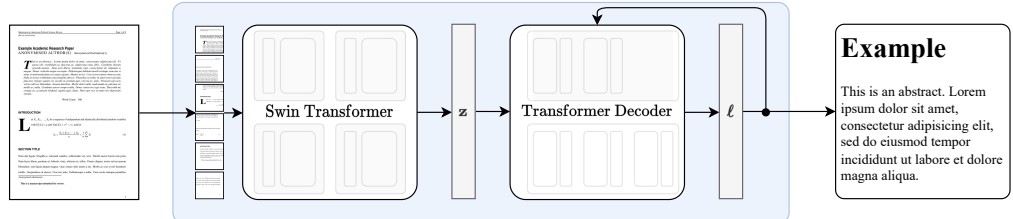

Figure 1: Our simple end-to-end architecture following Donut (Kim et al., 2022). The Swin Transformer encoder takes a document image and converts it into latent embeddings, which are subsequently converted to a sequence of tokens in a auto-regressive manner

and then padded to achieve the desired input size. This input size allows us to use the Swin base model architecture (Liu et al., 2021). We initialize the model with the pre-trained weights.

The Transformer decoder has a maximal sequence length of $S = 4096$. This relatively large sizing is due to the fact that the text of academic research papers can be dense and the syntax for tables in particular is token intensive. The BART decoder is a decoder-only transformer with 10 layers. The entire architecture has a total of 350M parameters.

We also test experiment with a smaller model (250M parameters) with a slightly smaller sequence length of $S = 3584$ and only 4 decoder layers, where we start from the pre-trained base model.

During inference the text is generated using greedy decoding.

**Training** We use an AdamW optimizer (Loshchilov & Hutter, 2019) to train for 3 epochs with an effective batch size of 192. Due to training instabilities, we choose a learning rate of $\text{lr}_{\text{init}} = 5 \cdot 10^{-5}$ which is reduced by a factor of 0.9996 every 15 updates until it reaches $\text{lr}_{\text{end}} = 7.5 \cdot 10^{-6}$.

## 3.2 DATA AUGMENTATION

In image recognition tasks, it is often beneficial to use data augmentation to improve generalization. Since we are only using digital-born academic research papers, we need to employ a number of transformations to simulate the imperfections and variability of scanned documents. These transformations include erosion, dilation, gaussian noise, gaussian blur, bitmap conversion, image compression, grid distortion and elastic transform (Simard et al., 2003). Each has a fixed probability of being applied to a given image. The transformations are implemented in the *Albumentations* (Buslaev et al., 2020) library. For an overview of the effect of each transformation, see Fig. A.1.

During training time, we also add perturbations to the ground truth text by randomly replacing tokens. We found this to reduce the collapse into a repeating loop significantly. For more details, see Section 5.4.

## 4 DATASETS

To the best of our knowledge there is no paired dataset of PDF pages and corresponding source code out there, so we created our own from the open access articles on arXiv.[3] For layout diversity we also include a subset of the *PubMed Central* [4] (PMC) open access non-commercial dataset. During the pretraining, a portion of the *Industry Documents Library* [5] (IDL) is included. See Table A.1 for the dataset composition.

**arXiv** We collected the source code and compiled PDFs from 1,748,201 articles released on arXiv. To ensure consistent formatting, we first process the source files using *LaTeXML*[6] and convert them into HTML5 files. This step was important as it standardized and removed ambiguity from the LaTeX source code, especially in mathematical expressions. The conversion process included replacing user-defined macros, standardizing whitespace, adding optional brackets, normalizing tables, and replacing references and citations with their correct numbers.

---

[3] https://arxiv.org/
[4] https://www.ncbi.nlm.nih.gov/pmc/
[5] https://www.industrydocuments.ucsf.edu/
[6] http://dlmf.nist.gov/LaTeXML/

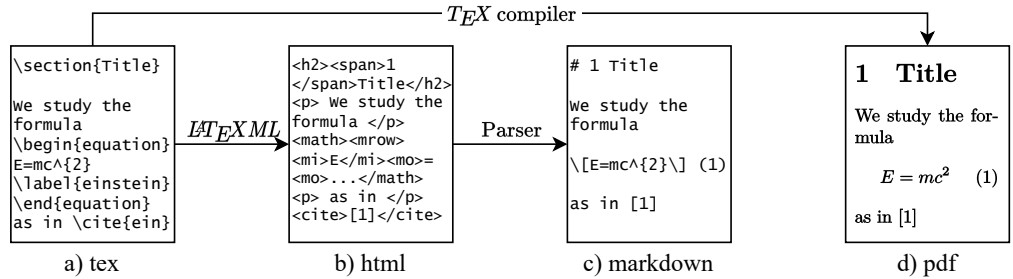

Figure 2: Data processing. The source file is converted into HTML which is then converted to Markdown. a) The LaTeX source provided by the authors. b) The HTML file computed form the LaTeX source using LaTeXML. c) The Markdown file parsed from the HTML file. d) The PDF file provided by the authors

We then parse the HTML files and convert them into a lightweight markup language that supports various elements such as headings, bold and italic text, algorithms, LaTeX inline and display math and LaTeX tables. This way, we ensure that the source code is properly formatted and ready for further processing.
The process is visualized in Fig. 2.

**PMC**     We also processed articles from PMC, where XML files with semantic information are available in addition to the PDF file. We parse these files into the same markup language format as the arXiv articles. We chose to use far fewer articles from PMC because the XML files are not always as rich in semantic information. Often times equations and tables are stored as images and these cases are not trivial to detect, which leads to our decision to limit the use of PMC articles to the pre-training phase.

The XML files are parsed into the same markup language as described above.

**IDL**     The IDL is a collection of documents produced by industries that have an impact on public health and is maintained by the University of California, San Francisco Library. Biten et al. (2022) provide high quality OCR text for PDFs from the IDL dataset. This does not include text formatting and is only used for pre-training to teach the model basic OCR of scanned documents.

## 4.1 SPLITTING THE PAGES

We split the markdown files according to the page breaks in the PDF file and rasterize each page as an image to create the final paired dataset. During the compilation, the LaTeX compiler determines the page breaks of the PDF file automatically. Since we are not recompiling the LaTeX sources for each paper, we must heuristically split the source file into parts, which correspond to different pages. To achieve that we are using the embedded text on the PDF page and match it to source text.
However, figures and tables in the PDF may not correspond to their position in the source code. To address this issue, we remove these elements in a pre-processing step using `pdffigures2` (Clark & Divvala, 2016). The recognized captions are are then compared to the captions in the XML file and matched based on their Levenshtein distance (Levenshtein, 1965). Once the source document has been split into individual pages, the removed figures and tables are reinserted at the end of each page. For a better matching we also replaced unicode characters in the PDF text with corresponding LaTeX commands using the pylatexenc-library[7].

**Bag of Words matching**     First we extract the text lines from the PDF using MuPDF[8] and preprocess them to remove page numbers and potential headers/footers. We then use a *Bag of Words* model (Harris, 1954) with TF-IDF vectorizer and a linear Support Vector Machine classifier. The model is fitted to the PDF lines with the page number as label. Next we split the LaTeX source into paragraphs and predict the page number for each of them.

---

[7]https://github.com/phfaist/pylatexenc
[8]https://mupdf.com/

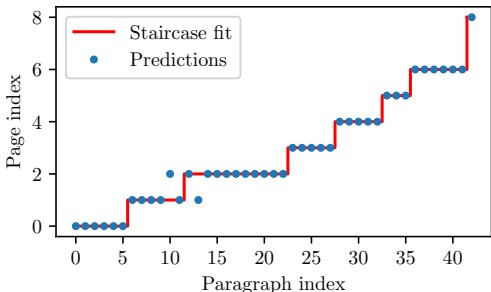

Figure 3: Example for splitting the paragraphs in the source code into different pages. The points in blue denote the page index predicted by the SVM.

Ideally, the predictions will form a stair case function but in practice the signal will be noisy. To find the best boundary points we employ a similar logic as decision trees and minimize a measure based on the *Gini* impurity

$$G_{[a,b]}(i) = (b-a) \cdot \left(1 - p_{[a,b]}^2(i) - p_{[a,b]}^2(i+1)\right), \tag{1}$$

where $p_{[a,b]}(i)$ is the probability of choosing an element with the predicted page number $i$ in the interval of paragraph indices $[a, b]$ that describes which paragraphs (elements) were considered for the split.

The best splitting position $t$ in the interval $[a, b]$ is then

$$\hat{t}_i = \arg \min_t \left(G_{[a,t]}(i) + G_{[t,b]}(i)\right). \tag{2}$$

The search process starts with all paragraphs and for each subsequent page break, the lower bound of the search interval is set to the previous split position. See Fig. 3 for a visualization of an example page.

**Fuzzy matching** After this first coarse document splitting we try to find the exact position within the paragraph. This is done by comparing the source text within the neighborhood of the predicted splitting position to the last sentences of the previous page of the embedded PDF text, and the first sentences of the next page using the `fuzzysearch` library[9]. If the two dividing points are at the same location in the source text, the page break is considered "accurate" and receives a score of 1. On the other hand, if the splitting positions differ, the one with the smallest normalized Levenshtein distance is selected and given a score of 1 minus the distance. To be included in the dataset, a PDF page must have an average score of at least 0.9 for both page breaks. This results in an acceptance rate of about $47\%$ of all pages.

## 4.2 GROUND TRUTH ARTIFACTS

Because the dataset was pre-processed by LaTeXML, the markup version of the source code can contain artifacts and commands from unsupported packages. The HTML file may contain subsection titles with numbering even though they are not numbered in the PDF. There may also be instances where figures or tables are missing from the ground truth due to processing errors.

In addition, the splitting algorithm of the source code will in some cases include text from the previous page or cut off words from the end. This is especially true for "invisible" characters used for formatting, like italic, bold text or section header.

For PMC papers the inline math is written as Unicode or italic text, while display math equations or tables are often included in image format and will therefore be ignored.

Each of these issues reduces the overall data quality. However, the large number of training samples compensates for these small errors.

---

[9]https://github.com/taleinat/fuzzysearch

# 5 RESULTS & EVALUATION

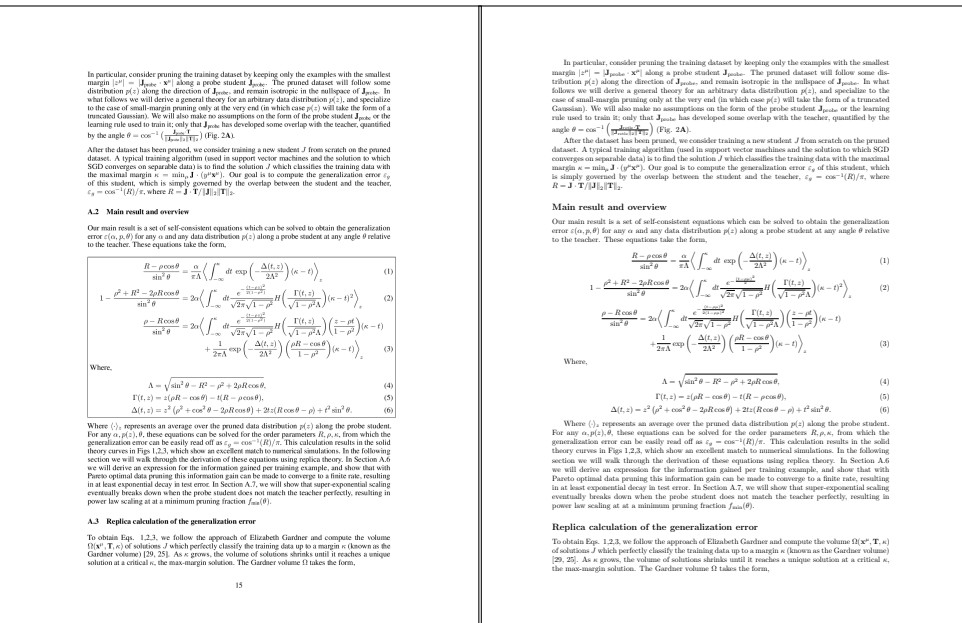

Figure 4: Example of a page with many mathematical equations taken from (Sorscher et al., 2022). Left: Image of a page in the document, Right: Model output converted to LaTeX and rendered to back into a PDF. Examples of scanned documents can be found in the appendix B.

In this section we discuss the results and performance of the model. For an example see Fig. 4 or go to Sec. B. The model focuses only on the important content relevant features of the page. The box around the equations is skipped.

## 5.1 METRICS

We report the following metrics on our test set.

**Character Error Rate** The character error rate (CER), or normalized Levenshtein distance (Levenshtein, 1965), measures the number of character manipulations (insertions, deletions, substitutions) it takes to get from one string to another.

**BLEU** The BLEU (Papineni et al., 2002) metric was originally introduced for measuring the quality of text that has been machine-translated from one language to another. The metric computes a score based on the number of matching n-grams between the candidate and reference sentence.

**METEOR** Another machine-translating metric with a focus on recall instead of precision, introduced in (Banerjee & Lavie, 2005).

**F-measure** We also compute the F1-score and report the precision and recall.

## 5.2 TEXT MODALITIES

In a scientific research article, there are three distinct types of text: 1) plain text, which comprises the majority of the document, 2) mathematical expressions, and 3) tables. It is important to separately examine each of these components during the evaluation process. This is necessary because in LaTeX, there are multiple ways to express the same mathematical expression. While some variability has been eliminated during the LaTeXML pre-processing step, there still is a significant amount of ambiguity present, like ordering of subscript and superscript, equivalent commands with different notation (`stackrel`, `atop`, `substack` or `frac`, `over`), situationally interchangeable commands (`bm`, `mathbf`, `boldsymbol`, `bf` or `\left(`, `\big(`, etc.), whitespace commands, additional layers of brackets, and more. As a consequence, there can be a discrepancy between prediction and ground

| Method | Modality | CER | BLEU | METEOR | Prec. | Rec. | F1 |
|---|---|---|---|---|---|---|---|
| PDF | All | 25.5 | 65.8 | 82.1 | 77.1 | 81.4 | 79.2 |
| GROBID | All | 31.2 | 55.6 | 71.9 | 74.0 | 72.1 | 73.0 |
| | Tables | 62.6 | 25.1 | 64.5 | 61.4 | 80.7 | 69.7 |
| + LaTeX OCR (Blecher) | Plain text | 36.3 | 57.4 | 69.2 | 82.1 | 70.5 | 75.9 |
| | Math | 72.7 | 0.3 | 5.0 | 11.0 | 8.6 | 9.7 |
| Nougat small | All | 7.3 | 88.9 | 92.8 | **93.6** | 92.2 | 92.9 |
| (250M*) | Tables | 22.0 | 68.5 | 78.6 | 75.0 | 79.8 | 77.3 |
| | Plain text | 5.8 | 91.0 | 94.3 | 96.1 | 95.3 | 95.7 |
| | Math | 11.7 | 56.0 | 74.7 | 77.1 | 76.8 | 76.9 |
| Nougat base | All | **7.1** | **89.1** | **93.0** | 93.5 | **92.8** | **93.1** |
| (350M*) | Tables | 21.1 | 69.7 | 79.1 | 75.4 | 80.7 | 78.0 |
| | Plain text | 5.8 | 91.2 | 94.6 | 96.2 | 95.3 | 95.7 |
| | Math | 12.8 | 56.9 | 75.4 | 76.5 | 76.6 | 76.5 |

Table 1: Results on arXiv test set. PDF is the text embedded in the PDF file. The modality "All" refers to the output text without any splitting. All metrics are reported in percent. *Number of parameters.

truth, even if the rendered formulas appear identical.

In addition, it is not always possible to determine, where an inline math environment ends and text begins, when writing numbers and punctuation (Example: `$\mathrm{H}_{0}$1`, vs. `H$_{0}1,$` $\rightarrow H_0 1$, vs. $H_0 1$,). This ambiguity reduces both math and plain text scores.

The expected score for mathematical expressions is lower than for plain text.

## 5.3 COMPARISON

We present our results in Table 1. As expected, the mathematical expressions have the worst agreement with the ground truth. For the plain text, most discrepancies come from formatting ambiguities and missing text due to inline math, as described above. The output format of GROBID is an XML file, which we convert into a compatible markup language, similar to the PMC or arXiv files. To some extent, GROBID provides support for formulas in its output, but it identifies and stores them as the Unicode representations embedded in the PDF. We replace each Unicode symbol with its corresponding LaTeX command to increase the similarity. Additionally, GROBID mislabels small inline expressions as text. For identified formulas, GROBID stores the bounding box coordinates. We modify the program by sending the snippet to the external formula recognition software LaTeX-OCR (Blecher, 2023). This way we can also get a signal for math modality. The reported results in this section are quite poor, primarily due to the amount of missed formulas by GROBID and the equation prediction accuracy is affected by the quality of the bounding boxes. The performance of the embedded PDF text alone is better than GROBID, which is due to formatting differences for the title page or reference section.

Both Nougat small and base are able to outperform the other approach and achieve high scores in all metrics. We note that the performance of the smaller model is on par with the larger base model.

## 5.4 REPETITIONS DURING INFERENCE

We notice that the model degenerates into repeating the same sentence continuously, a state from which it cannot autonomously recover. In its simplest form, the last sentence or paragraph is repeated ad infinitum. We observed this behavior in $1.5\%$ of pages in the test set, but the frequency increases for out-of-domain documents. Our evaluation extends to a diverse selection of academic documents, encompassing a range of sources such as scanned books. See Appendix B for examples.

Getting stuck in a repetitive loop is a known problem with Transformer-based models, when sampled with greedy decoding (Holtzman et al., 2020). We conducted experiments with nucleus sampling,

but encountered a susceptibility to repetitive outputs. Consequently, we opted against utilizing this technique due to the introduced randomness failing to effectively mitigate the issue of unwanted sentence repetitions.

It can also happen that the model alternates between two sentences but sometimes changes some words, so a strict repetition detection will not suffice. Even harder to detect are predictions where the model counts its own repetitions, which sometimes happens in the references section.

In general we notice this kind behavior after a mistake by the model. The model is not able to recover from the collapse.

**Anti-repetition augmentation**    Because of that we introduce a random perturbation during training. This helps the model to learn how to handle a wrongly predicted token. For each training example, there is a fixed probability that a random token will be replaced by any other randomly chosen token. This process continues until the newly sampled number is greater than a specified threshold (in this case, 10%). We did not observe a decrease in performance with this approach, but we did notice a significant reduction in repetitions. Particularly for out-of-domain documents, where we saw a 32% decline in failed page conversions.

**Repetition detection**    Since we are generating a maximum of 4096 tokens, the model will stop at some point. However, it is very inefficient and resource-intensive to wait for an "end of sentence" token that will not be generated. To detect repetition during inference, we examine the largest logit value $\ell_i = \max \boldsymbol{\ell}_i$ for the ith token. We observed that logits after a collapse can be separated using the following heuristic consisting of two steps. First, calculate the variance of the logits for a sliding window of size $B = 15$

$$\mathrm{VarWin}_B[\boldsymbol{\ell}](x) = \frac{1}{B} \sum_{i=x}^{x+B} \left( \ell_i - \frac{1}{B} \sum_{j=x}^{x+B} \ell_j \right)^2 . \tag{3}$$

Here, $\ell$ is the signal of logits, and $x$ is the index. Using this new signal, compute variances again, but this time from point $x$ to the end of the sequence:

$$\mathrm{VarEnd}_B[\boldsymbol{\ell}](x) = \frac{1}{S-x} \sum_{i=x}^{S} \left( \mathrm{VarWin}_B[\boldsymbol{\ell}](i) - \frac{1}{S-x} \sum_{j=x}^{S} \mathrm{VarWin}_B[\boldsymbol{\ell}](i) \right)^2 . \tag{4}$$

If this signal drops below a certain threshold (we choose 6.75) and stays below for the remainder of the sequence, we classify the sequence as having repetitions.

During inference time, it is not possible to compute the entire sequence if our goal is to stop generation earlier. Here, we work with a subset of the last 200 tokens and half the threshold. After the generation is finished, the procedure as described above is repeated for the full sequence.

## 5.5 LIMITATIONS & FUTURE WORK

**Utility**    The utility of the model is limited by a number of factors. First, the problem with repetitions outlined in section 5.4. The model is trained on research papers, which means it works particularly well on documents with a similar structure. However, it can still accurately convert other types of documents.

Nearly every dataset sample is in English. Initial tests on a small sample suggest that the model's performance with other Latin-based languages is satisfactory, although any special characters from these languages will be replaced with the closest equivalent from the Latin alphabet. Non-Latin script languages result in instant repetitions.

**Generation Speed**    On a machine with a NVIDIA A10G graphics card with 24GB VRAM we can process 6 pages in parallel. The generation speed depends heavily on the amount of text on any given page. With an average number of tokens of $\approx 1400$ we get an mean generation time of 19.5s per batch for the base model without any inference optimization. Compared to classical approaches (GROBID 10.6 PDF/s (Lopez, 2023)) this is very slow, but it is not limited to digital-born PDFs and can correctly parse mathematical expressions.

**Future work**    The model is trained on one page at a time without knowledge about other pages in the document. This results in inconsistencies across the document. Most notably in the bibliography where the model was trained on different styles or section titles where sometimes numbers are skipped

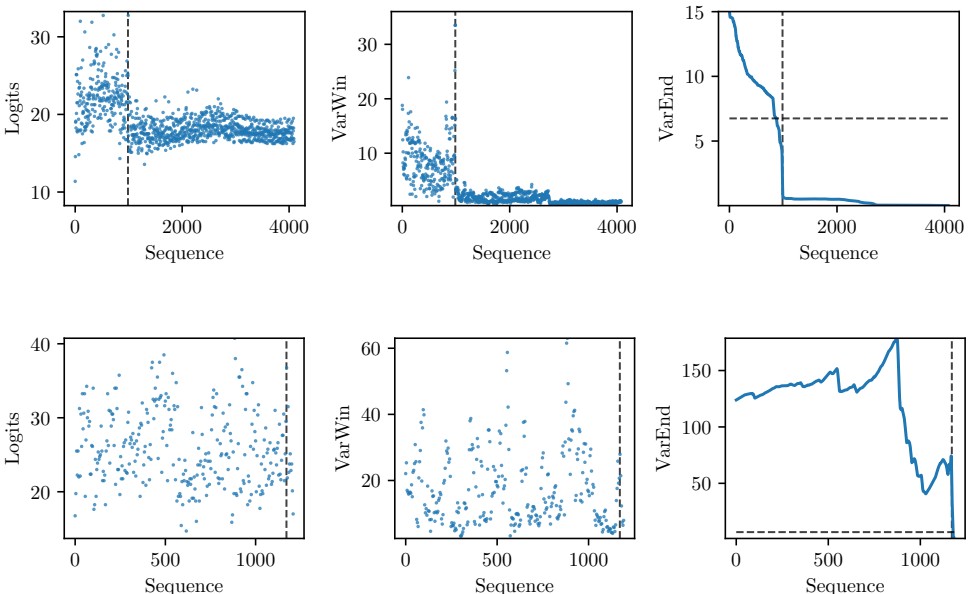

Figure 5: Examples for repetition detection on logits. The vertical index denotes the sequence index, where repetitions start. Top: Sample generation with repetition, Bottom: Sample generation without repetition. Left: Highest logit score for each token in the sequence $\ell(x)$, Center: Sliding window variance of the logits $\mathrm{VarWin}_B[\ell](x)$, Right: Variance of the window variance (3) from the position to the end $\mathrm{VarEnd}_B[\ell](x)$ with threshold as horizontal dashed line.

or hallucinated. Though handling each page separately significantly improves parallelization and scalability, it may diminish the quality of the merged document text.

The primary challenge to solve is the tendency for the model to collapse into a repeating loop, which is left for future work.

## 6 CONCLUSION

In this work, we present Nougat, an end-to-end trainable encoder-decoder transformer based model for converting document pages to markup. We apply recent advances in visual document understanding to a novel OCR task. Distinct from related approaches, our method does not rely on OCR or embedded text representations, instead relying solely on the rasterized document page. Moreover, we have illustrated an automatic and unsupervised dataset generation process that we used to successfully train the model for scientific document to markup conversion. Overall, our approach has shown great potential for not only extracting text from digital-born PDFs but also for converting scanned papers and textbooks. We hope this work can be a starting point for future research in related domains.

All the code for model evaluation, training and dataset generation can be accessed at `https://github.com/facebookresearch/nougat`.

### ACKNOWLEDGMENTS

Thanks to Ross Taylor, Marcin Kardas, Iliyan Zarov, Kevin Stone, Jian Xiang Kuan, Andrew Poulton and Hugo Touvron for their valuable discussions and feedback.

Thanks to Faisal Azhar for the support throughout the project.

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

# A  DATASET

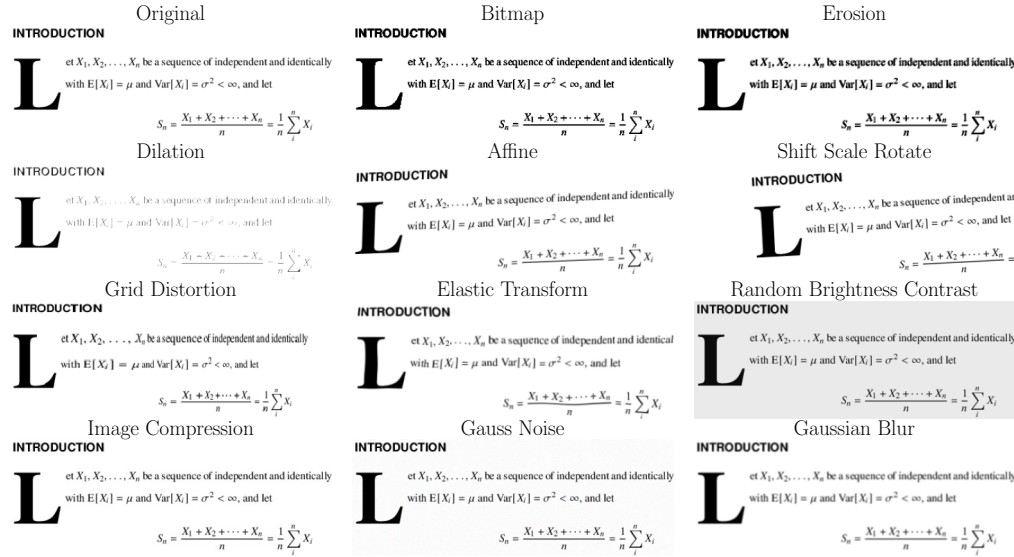

Figure A.1: List of the different image augmentation methods used during training on an example snippet form a sample document.

| Name  | Number of Pages |
|-------|-----------------|
| arXiv | 7,511,745       |
| PMC   | 536,319         |
| IDL   | 446,777         |
| **Total** | **8,494,841** |

Table A.1: Dataset composition

The most important data source is arXiv, making up $> 88.4\%$ of the corpus. On arXiv most research documents are paired with the LaTeX source code provided by the authors. The LaTeX source offers more information and is left unprocessed, unlike the XML format from PMC where equations and tables are frequently substituted with images. This allows us to select exactly which information we need to build the dataset.

# B  EXAMPLES

In this section we converted some pages from old text books using the Nougat base model. The text books from the *Internet Archive*[10] and *Project Gutenberg*[11] and are in public domain.
The performance for these scanned pages is noticeable worse than for digital-born documents. However, the model does generate sensible text for each page with few errors. For example see the first row of Fig. B.1. Here the model mistakes the almost illegible exponent $n$ for $*$. In the second row of the same figure the model falls into a repetitive loop after predicting another comma instead of a dot. Similar problems can be seen in Fig. B.2.
In Fig. B.3 we present pages, scanned with a mobile device, from a printed master thesis and the Nougat output. The model is robust to the artifacts that arise when hand-scanning a document.
Explore the examples in this section on the project page: `https://facebookresearch.github.io/nougat/`.

---

[10]`https://archive.org/`
[11]`https://www.gutenberg.org/`

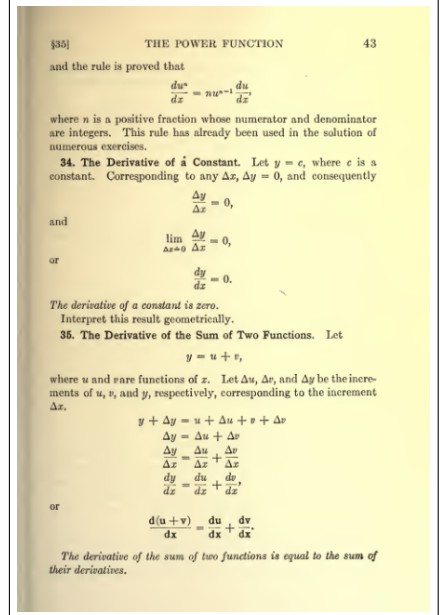
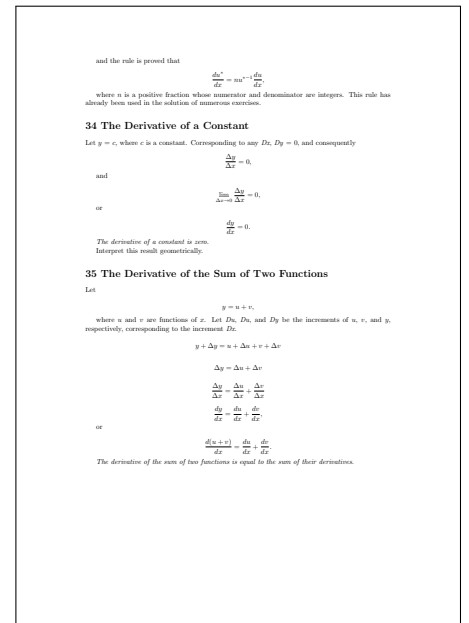
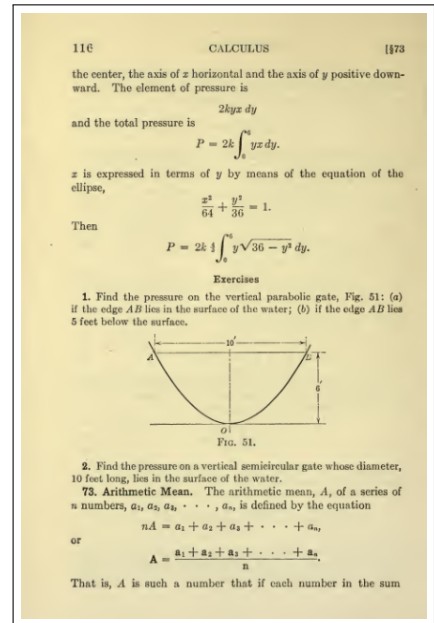
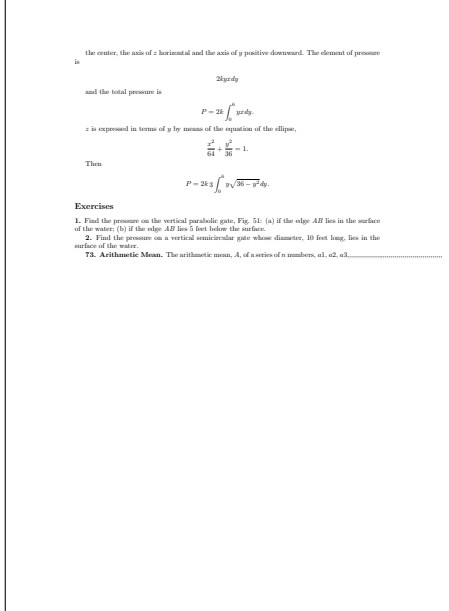

Figure B.1: Example of an old calculus text book (March & Wolff, 1917).

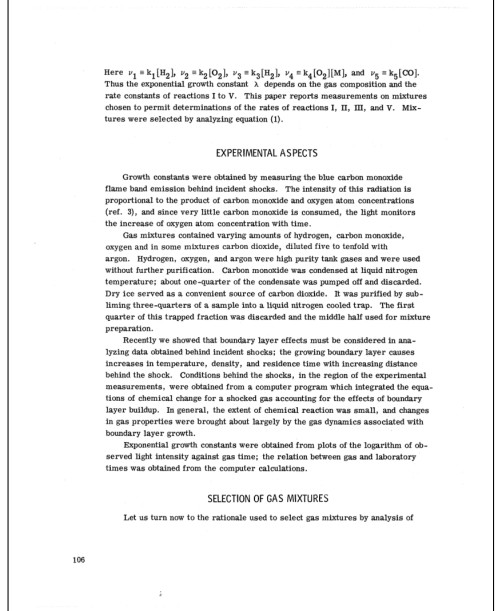

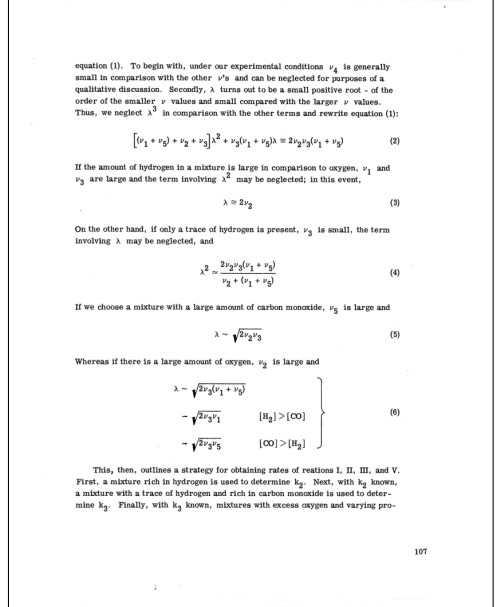

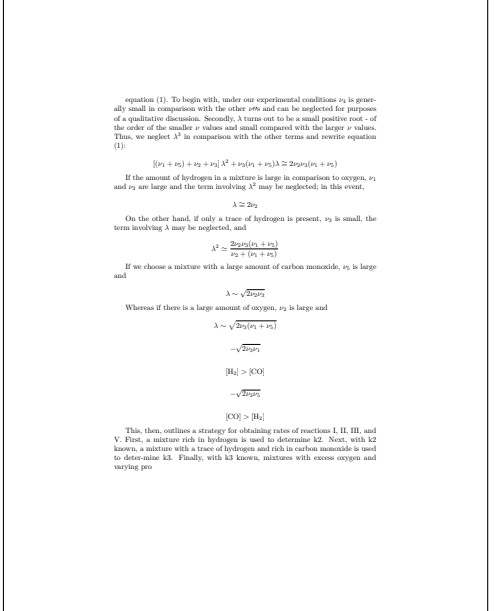

Figure B.2: A selection of pages from a NASA conference from 1970 (Gordon et al., 1970).

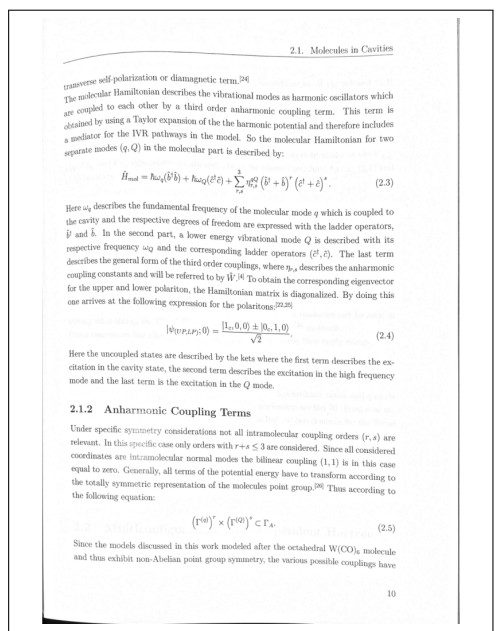
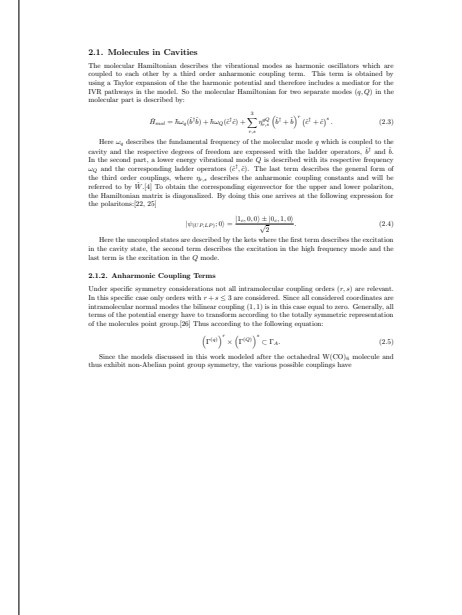
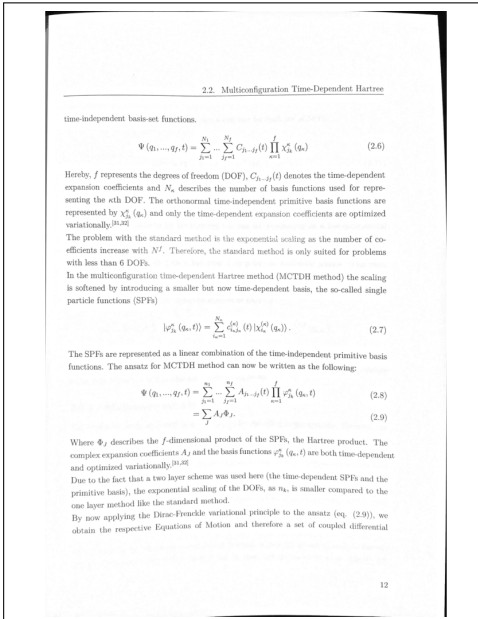
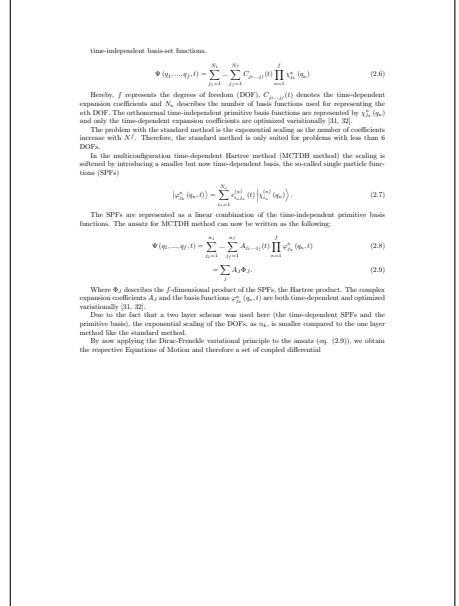

Figure B.3: Scan of a modern thesis with a mobile device camera, with permission from the author.

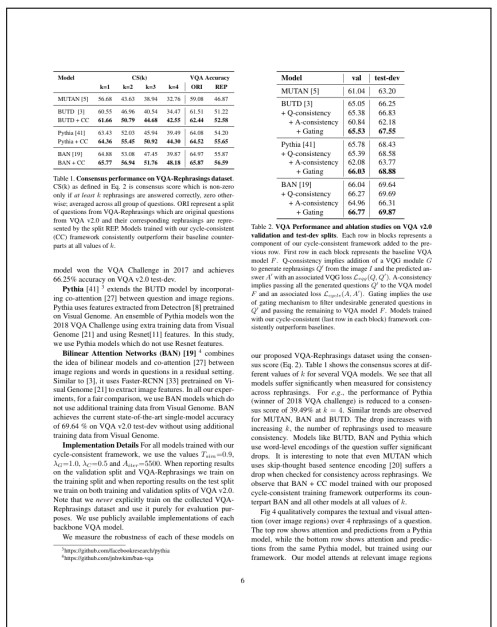
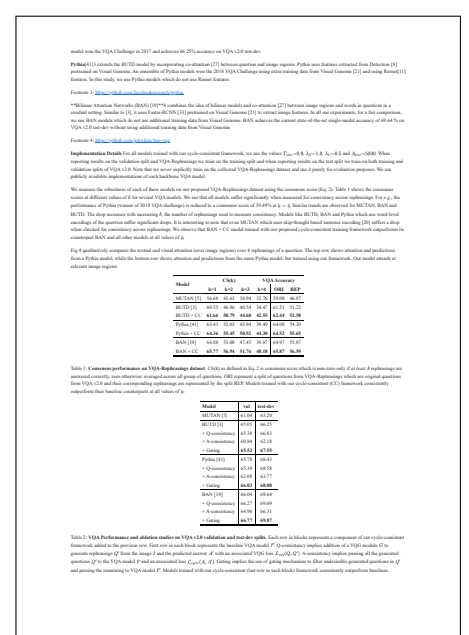
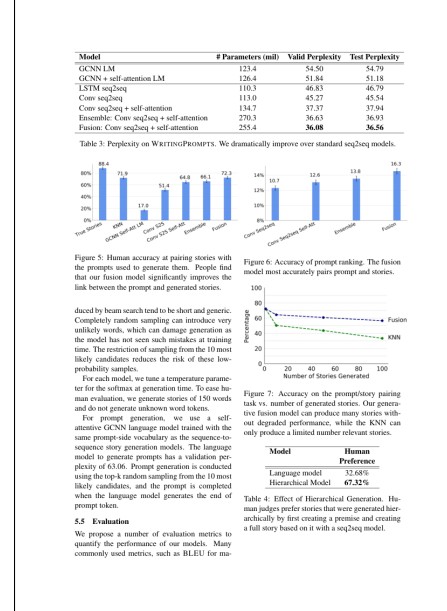

Figure B.4: Pages with tables. Upper: Fan et al. (2018) page 6, Lower: Shah et al. (2019) page 6

