# OpenReview forum: "Nougat: Neural Optical Understanding for Academic Documents"
_ICLR.cc/2024/Conference — ICLR 2024 poster_

### Official Review · Reviewer_YnWx · 2023-10-26

**Soundness:** 3 good
**Presentation:** 2 fair
**Contribution:** 4 excellent
**Rating:** 5
**Confidence:** 4

**Summary:**

This article presents an end-to-end model for converting PDF image files into a markup language, with a focus on accurately reconstructing mathematical formulas. The authors also discuss the preprocessing pipeline used to align PDFs with LaTeX source code and provide details on the datasets used. The model can be applied to other types of scanned document images.

**Strengths:**

1. The proposed model significantly improves the processing of scientific documents into a markup language.

2. The model performs exceptionally well in parsing mathematical formulas, surpassing algorithm-based OCR engines.

3. The method is simple yet effective.

**Weaknesses:**

1. The lack of novelty in the model architecture’s methodology is a weakness. It suggests that the approach may not bring any new or innovative ideas to the field. It would be better for an industrial track paper.

2. The reported results being limited to English data is a weakness. It is unclear if the model can support other languages, such as Chinese or Japanese. However, it would be interesting to evaluate the transfer learning ability of the model for layout analysis, which could potentially address this limitation.

3. Figure 5 and section 5.4 on repetitions during inference being difficult to follow. It indicates that the explanation or presentation of this aspect of the model may not be clear or well-explained, making it challenging for readers to grasp the concept.

**Questions:**

1. What does “out-of-domain documents” refer to on page 8, and what is the objective of conducting experiments on repetition data?

2. In Eq. X, does “a” or “b” denote a paragraph index?

3. Why were Table, Plain text, and Math treated as separate modalities for experimentation, and how does this differ from conventional methods?

4. Can the model be directly applied to visually rich document understanding tasks?

5. Would it be better to move the first paragraph in Section 3 Model from page 2 to the related work section?


Other suggestions:  give eq. no and full stops

---

> ### Author Response · Authors · 2023-11-22
>
> Thank you for your thoughtful feedback. We appreciate your diligence in reviewing our work.
> ## Weaknesses
> > The lack of novelty in the model architecture's methodology is a weakness. It suggests that the approach may not bring any new or innovative ideas to the field. It would be better for an industrial track paper.
>
> Please refer to our general response 1.
> > The reported results being limited to English data is a weakness. It is unclear if the model can support other languages, such as Chinese or Japanese. However, it would be interesting to evaluate the transfer learning ability of the model for layout analysis, which could potentially address this limitation.
>
>
> Please refer to our general response 2.
> > Figure 5 and section 5.4 on repetitions during inference being difficult to follow. It indicates that the explanation or presentation of this aspect of the model may not be clear or well-explained, making it challenging for readers to grasp the concept.
>
> Thank you for your feedback. We will rework this section to make it easier to follow.
>
> ## Questions
> > What does “out-of-domain documents” refer to on page 8, and what is the objective of conducting experiments on repetition data?
>
> “out-of-domain documents” are in our case academic documents found on the internet including scanned or digital books (for examples, see in Appendix B: Fig B.1 and Fig B.2). We realize this was not clear in the original text and updated it. Thank you for the pointer.
> > In Eq. X, does “a” or “b” denote a paragraph index?
>
> Yes, in Eq. X, “a” or “b” denote a paragraph index.
> > Why were Table, Plain text, and Math treated as separate modalities for experimentation, and how does this differ from conventional methods?
>
> In our study, we chose to categorize Tables, Math, and Plain Text as distinct text modalities, only for our evaluation process. This distinction was made to gain more detailed insights into the potentially varying results, acknowledging their inherent differences. For instance, the way Tables might yield different outcomes compared to Plain Text is noteworthy. It might be more apt to use the term "category" rather than "modality". Importantly, this differentiation was applied only in the evaluation phase, not during training. As such, the reference to 'conventional methods' isn't directly applicable here, since our approach didn't alter the training methodology, but rather focused on a more nuanced evaluation strategy.
>
> > Can the model be directly applied to visually rich document understanding tasks?
>
> Please refer to our general response 3.
> > Would it be better to move the first paragraph in Section 3 Model from page 2 to the related work section?
>
> Yes, that would be better, thank you for the suggestion!

---

### Official Review · Reviewer_jN3C · 2023-10-30

**Soundness:** 2 fair
**Presentation:** 3 good
**Contribution:** 1 poor
**Rating:** 3
**Confidence:** 5

**Summary:**

This paper presents a way to build a system that converts an image of a scientific document to its Markup representation from both modeling and data collection perspectives. The use of Markup as its representation enables to handle texts, math equations, tables uniformly by a single model. The model is a basic end-to-end transformer-based autoregressive model. In order to mitigate the common repetition problem of such models, a simple data augmentation method is proposed. It also describes the data construction procedure to create image-Markdown pairs from arXiv where several practical techniques are required. The experimental results show that the transformer-based model is capable for the task and outperforms a baseline system. The code and pre-trained model will be released on GitHub.

**Strengths:**

I see there are mainly three contributions in the paper:

1. It proposes to use Markdown to extract structure information from images of scientific documents.

The use of such representations (e.g. LaTex, html, etc.) in the OCR domain is not novel (e.g. math equation recognition has been studied actively in the community for many years), but proposing to use such representations for this particular domain and tackling the problem seriously should get some credit. If scientific documents stored as images can be converted to Markdown, there should be great values to humanity.

2. It describes the data construction procedure for the task.

Given the capability of latest ML models, the most challenging part is to find data to train models. Identifying arXiv as a good source for the problem and providing practical procedure to construct a dataset seems to have good values to subsequent studies.

3. It shows a transformer-based end-to-end model works for the task and a simple technique further improves the accuracy.

It is not very surprising that the Donut-based model works for the in-domain data, but it should have a certain value to show that it works well in the setting. The proposed data augmentation is simple but effective, and good analyses are given.

**Weaknesses:**

I am not very certain about its scientific contributions to the ML community.

In a sense, this paper narrows down the large problem space of OCR into the limited domain and proposes a specialized solution to it. There is definitely a practical value if we can extract Markdown for scientific documents and it is important to have a technical solution to the problem. However, it is not really surprising that a transformer-based model can do a reasonable job for the task. In my opinion, it is challenging to claim novelty for the adoption of transformer for the task and by releasing code and pre-trained models. In my view, the novelty could be claimed for the problem definition and the data construction procedure. Both may be valuable in a certain field, but I am not very sure how much they are for a top-tier general ML conference. I see a lot of high-quality work in the paper, but I feel ICLR is not the best destination for the paper.

**Questions:**

I think the important details are well described and there is no particular question.

**Details Of Ethics Concerns:**

No concern.

---

> ### Author Response · Authors · 2023-11-22
>
> Thank you for your thoughtful feedback. We appreciate your diligence in reviewing our work, however we disagree regarding the lack of novelty, please see our general response 1.

---

> > ### Comment · Reviewer_jN3C · 2023-11-23
> > **Thank you for the response**
> >
> > Thank you for the rebuttal. I'm going to have a discussion with other reviewers and make a final proposal.

---

### Official Review · Reviewer_SnU5 · 2023-10-30

**Soundness:** 3 good
**Presentation:** 3 good
**Contribution:** 3 good
**Rating:** 8
**Confidence:** 5

**Summary:**

This paper introduces a new model (Nougat) for automatic understanding of academic documents.
The authors highlight the main limitations of SOTA methods in the conversion of scientific articles, mainly in dealing with complex objects like tables & mathematical equations. As a result, Nougat addresses end-to-end scientific article understanding with a focus on math equations. The authors develop a large dataset of scientific articles and train Nougat to predict their Markdown formatting. Results show that Nougat outperforms SOTA approaches by a large margin. Examples of article conversion are also presented in the appendix.

**Strengths:**

- The experimental parameters are meticulously described in the article, with extensive information covering preprocessing, model architecture, and training parameters. Notably, the authors have made both the code and the model available on GitHub, which strengthens the reproducibility of their work. The level of detail in their descriptions provides confidence in the potential to replicate these results.

- The methodology used to create the training dataset is also well described and will be available on GitHub. Scientific articles from three sources (ArXiv, PubMed, and IDL) are used. The process of automatic conversion from LaTeX to Markdown is explained step by step, with particular emphasis on the quality control measures taken. The appendix describes the dataset in detail.

- Nougat outperforms GROBID by a substantial margin in the recognition of plain text, tables, and mathematical expressions. Notably, the significant improvement in handling mathematical equations is particularly impressive, given that this was a significant weakness of GROBID.

**Weaknesses:**

- Nougat outperforms GROBID by a substantial margin in the recognition of plain text, tables, and mathematical expressions. However, I feel that the results section is lacking a qualitative comparison between GROBID, Nougat-base and Nougat-small, especially on mathematical expressions.

- The GROBID system is not sufficiently described, and authors should comment its performance on plain text (what is LaTeX OCR?)

- Two Nougat models are compared in the paper (base and small). However, the article does not provide a comprehensive comparison of these architectures (performance gain, training and inference times, energy consumption…). Consequently, it leaves a significant question unanswered: whether it’s justifiable to opt for the base model when the small model performs almost as well..

- I found the authors’ perspective on how to deal with repetitions very valuable, as this is a well-recognized challenge when working with transformer models. However, the gain should be measured for the repetition detection module, especially since the threshold is set manually and would be painful to adapt to other models/applications.

Finally, some limitations of this work are highlighted, although I feel that this section could be developed further.

**Questions:**

* Are the scores presented in % for the editdistance metric? This metric is referred to as CER (Character Error Rate) in the document understanding community.
* This sentence is not clear, as Fig. 2 shows that LaTeX sources are recompiled. In Overleaf you can match paragraphs in the LaTeX source and their localization in the PDF - can this feature be used to simplify this step?
> Since we are not recompiling the LaTeX sources for each paper, we must heuristically split the source file into parts, which correspond to different pages. To achieve that we are using the embedded text on the PDF page and match it to source text.
* The label noise strategy is not clear. Is noise added to 10% of labels?
> This process continues until the newly sampled number is greater than a specified threshold (in this case, 10%)
* Text vs equations vs tables all have different semantics. Could you comment on which metrics are the most relevant for each text category? It would be interesting to use specific metrics for equation recognition (see the CHROME competition https://www.cs.rit.edu/~rlaz/files/CROHME+TFD%E2%80%932019.pdf)
* The limitation of 4096 tokens seems low for a full page. Did you encounter pages with a larger number of tokens to predict, and how did you handle them? How would this limitation affect document-level recognition (sliding window/patch?).
* Is any post-processing strategy used to normalize section/subsection formatting (example: `##` (section) predicted instead of `###` (subsection)) ?
* Adaptability to other languages? While most publications are written in English, some of them are written in other languages. How much work to fine-tune Nougat? (since the tokenizer, dataset, pre-trained models are all specialized for English)
* What if repetitions appear inside the page and not at the end? (ex: a sentence is repeated twice, but after that the model continues reading until the end of the page). In this case your method will not work as it will just stop the generation. 1) does this case happen? 2) how would you deal with this issue?
* Do you handle references to other parts of the page? (Ref to tables, figures in the text?). It might not be crucial when dealing with pages, but it will be useful at document-level.
* Do you handle references to the original image/document (e.g. select a word in the Markdown to highlight it in the PDF)? This feature could be used by users to check an equation if it does not make any sense, for example.
* To go beyond recognition, how could Nougat be adapted to a multitask setting with e.g. Visual Question Answering / Neural Machine Translation / Summarization? It would certainly be useful to allow users to ask questions about the article: ask for an explanation, ask for a reference, ask to translate or summarize...

Errors/typo
* Table A.1. The total does not add up to the number of pages for each source (should be 8,494,841). As a consequence, % for each source are also wrong.
* Typo p9 (last paragraph of section 5.4) "to compute the to the end "

---

> ### Author Response · Authors · 2023-11-22
>
> Thank you for the thorough review. We agree with your assessments and added more context regarding GROBID in the paper. In Section 5.3, we provide a clarification on the utilization of LaTeX OCR. That might not have been apparent solely from the table, so we included a reference there as well.
>
> > Are the scores presented in % for the editdistance metric? This metric is referred to as CER (Character Error Rate) in the document understanding community.
>
> Thank you for the pointer. The scores are not presented in %. We will rename the metric in the paper.
> > Fig. 2 shows that LaTeX sources are recompiled
>
> Recompiling LaTeX sources using SyncTeX could indeed streamline our data generation process, but it comes at the cost of significantly increased computational requirements.
> To mitigate this, we have opted for a more general approach by utilizing the original PDF files supplied by the authors. This choice allows us to accommodate various data sources, including cases such as PMC documents where no LaTeX source files are provided.
> Fig. 2 was only supposed to show the relationship between the tex and pdf files, not to imply that we are recompiling the LaTeX sources for each paper. The image caption does clarify the relationship: “d) the PDF file provided by the authors”
> > The label noise strategy is not clear
>
> Thank you for the feedback. The strategy is as follows:
> With a chance of 10% we change one token in the ground truth text to another randomly selected token. The likelihood of changing two tokens simultaneously is 1%, calculated as (10%)^2. Similarly, the probability diminishes exponentially with each additional token substitution, such that the chance of changing three tokens is 0.1%, and so forth.
> > Text vs equations vs tables all have different semantics. Could you comment on which metrics are the most relevant for each text category?
>
> In my opinion, the CER is the most important metric for all of the modalities, since we pre-process all equations and tables to reduce ambiguity. However, using a symbolic graph could help give a clearer signal. Thank you for the pointer, we will investigate it in the future.
> > The limitation of 4096 tokens seems low for a full page
>
> On average a page has 1400 tokens. The 4k context size is more than enough for most of the pages. The only problem we noticed are with large tables, because they are token intensive since the GALACTICA tokenizer was not trained on LaTeX tables.
> We will leave long context exploration to future work.
> > post-processing strategy used to normalize section/subsection formatting
>
> No, we don’t have a post-processing strategy for section titles.
> > Adaptability to other languages
>
> Please refer to our general response 2.
> > What if repetitions appear inside the page and not at the end? (ex: a sentence is repeated twice, but after that the model continues reading until the end of the page). In this case your method will not work as it will just stop the generation. 1) does this case happen? 2) how would you deal with this issue?
>
> If a sentence is repeated in the document, the model is able to do that as well and continue with the rest of the text. The repetition detection during inference will not stop the generation, because the logits variance does not drop because of it.
> > Do you handle references to other parts of the page
> All references (to e.g. citations, formulas, tables, figures) are replaced with the corresponding label in the LatexML pre-processing step. So you would need to pair them up again during post-processing.
> > Do you handle references to the original image/document (e.g. select a word in the Markdown to highlight it in the PDF)? This feature could be used by users to check an equation if it does not make any sense, for example.
> > To go beyond recognition, how could Nougat be adapted to a multitask setting with e.g. Visual Question Answering / Neural Machine Translation / Summarization?
>
> This is a good idea, but we did not look into it yet. One could possibly use the attention maps of the vision transformer to highlight the most important image sections for a given output token.
> We agree, adapting Nougat to Visual Question Answering would be an interesting direction. Please refer to our general response 3.

---

### Official Review · Reviewer_VwBw · 2023-11-03

**Soundness:** 3 good
**Presentation:** 3 good
**Contribution:** 4 excellent
**Rating:** 10
**Confidence:** 4

**Summary:**

The paper proposes an image-to-LaTeX model for converting PDF documents into the corresponding LaTeX source. The model comprises of Swin Transformer encoder, followed by a standard text decoder (BART). The paper also describes in detail an involved process of collecting the data to obtain the training dataset, which involves non-trivial steps to clean and split PDFs. The model and the code to generate the dataset are made available, and the model outperforms several existing benchmarks.

**Strengths:**

Significance: The paper has an excellent contribution, first and foremost by creating the dataset for pdf-2-latex and describing the methodology and individual steps behind it - these two will greatly benefit the community. The availability of the model is also a significant contribution.
Originality: The proposed modelling approach is not original, but the construction of the dataset is.
Quality & Clarity: The paper is well written and clear, and reproducibility is further fostered by the model / data generation release.

**Weaknesses:**

No significant weaknesses.

**Questions:**

One of the central topics in the results is about the repetition of the same sentence again, an artifact of the greedy sampling, and authors approach this problem by doing a data augmentation to help reduce those repetitions. Is there a particular reason why the authors chose to stick with greedy sampling instead of using Top-K / Top-p sampling, which is a standard approach to reduce the overconfidence during autoregressive decoding? (See ex. "The curious case of neural text degeneration")

---

> ### Author Response · Authors · 2023-11-22
>
> Thank you for your thoughtful feedback.
>
> In our OCR problem domain, the task is to predict the next token, and there is only one correct token to be predicted at each step. Given this nature of the task, we opted for greedy decoding as it ensures determinism in our predictions. Nucleus sampling introduces a level of randomness that is unnecessary for our specific use case, as there is a single correct token at each decoding step.
>
> During the course of our experimentation, we did explore other sampling techniques such as top-k and top-p. However, we observed that these methods were also prone to introducing repetitions in the generated sequences. In the next version of our paper, we will report those additional observations.

---

### Author Response · Authors · 2023-11-22
**General Response**

# General Response 1 - Lack of Novelty
Both jN3C and YnWx raised similar concerns about the lack of novelty in the model architecture’s methodology:

jN3C: “However, it is not really surprising that a transformer-based model can do a reasonable job for the task. In my opinion, it is challenging to claim novelty for the adoption of a transformer for the task and by releasing code and pre-trained models.“

YnWx: “The lack of novelty in the model architecture’s methodology is a weakness. It suggests that the approach may not bring any new or innovative ideas to the field.”

While indeed we did not proposed a new architecture, we argue that our paper has important novelty:
- Data methodology: the construction of the dataset is original, as acknowledged by reviewers  VwBw and SnU5. It involves a complex pipeline from the data collection, the cleaning and split of the PDFs, some data augmentation, and the data construction procedure to create image-markdown pairs.
- Results: our model outperforms GROBID and obtains a new state-of-the-art. The novelty of our work lies in the application and optimization of this architecture for our specific task. Our approach has been meticulously tailored to address unique challenges in this domain, which were not adequately solved by existing models.
- Longer term impact: In the context of Large Language Models, high quality data is known to be the most important factor. By focusing on making scientific data more accessible and usable, our work contributes significantly to the field of LLMs. This approach is not just about current results; it sets a framework for how AI can better handle complex academic information in the future. Our commitment to sharing our code, model, and data generation methods ensures that others can reproduce and build upon our work. This has the potential to aid in various scientific areas. Note the resulting model has only 350M parameters, making it usable at scale in practice.

# General Response  2 - Multilingual Nougat
Both reviewers SnU5 and YnWx asked regarding the adaptability to other languages? It would have great applications, and while we’ll leave the development of a multilingual version for future work, here some elements discussing this question, including evidences of some multilingual generalization of our current Nougat:

- As stated in Section 5.5, for Latin script languages we did some qualitative tests, indicating that in fact Nougat generalizes well, despite being trained on an English corpus. Note that it would still drop some special characters. Here is a NOT cherry picked example of a Polish paper:
  Original PDF text:
  ```
  Teoria ekonomii oraz wynikające z niej zasady polityki gospodarczej notuje w swej historii zasadnicze zwroty, które zwykle są ściśle związane z tym, co się dzieje w gospodarce. Keynesowska rewolucja była odpowiedzią na wielką depresję z lat 1929–1933, której przebiegu nie można było wytłumaczyć na gruncie ekonomii klasycznej. Sam keynesizm wyczerpał swoje możliwości w obliczu stagflacji z przełomu lat 70. i 80. XX wieku, zaczęto więc poszukiwać nowych podstaw teoretycznych dla polityki gospodarczej
  ```
  Nougat prediction:
  ```
  Teoria ekonomii oraz wynikajace z niej zasady polityki gospodarczej notuje w swej historii zasadnicze zwroty, ktore zwylke sa scisle zwiazane z tym, co sie dzieje w gospodarce. Keynesowska rewolucja byla odpowiedzia na wielka depresje z lat 1929-1933, ktorej przebiegu nie mozna bylo wythumaczyc na gruncie ekonomii klasycznej. Sam keynesizm wyczerpal swoje mozliwosci w obliczu stagflacji z przelomu lat 70. i 80. XX wieku, zaczeto wiec poszukiwac nowych podstaw teoretycznych dla polityki gospodarczej
  ```
  Link to the Polish paper: https://cejsh.icm.edu.pl/cejsh/element/bwmeta1.element.desklight-368c7595-0ce3-4ba8-8e8c-0fd90c229d29
- For non-latin script languages e.g. Japanese or Chinese would not work at all as it is, without a multilingual adaptation.
- Research in this direction makes us optimistic given that (i) Nougat contribution and method is primarily data-oriented, and so it would only be about collecting multilingual sources of data or using existing ones (e.g. https://arxiv.org/abs/2301.11312). Moreover, the community has now established strong techniques to tackle multilinguality, even including the possibility to transfer monolingual representations to a multilingual setup (see https://arxiv.org/abs/1910.11856), which may enable to adapt Nougat, without the need of retraining it.

# General Response 3 - Visual Tasks
Both reviewers SnU5 and YnWx asked about going beyond the recognition task to visual understanding tasks. The underlying architecture is derived from Donut. In the paper, the authors show that their model can successfully be finetuned on multiple different visual understanding tasks, such as information extraction and question answering. Experiments will be needed to see how well our model can adapt to these new tasks.

---

### Meta-Review · Area_Chair_iEq9 · 2023-12-19

**Metareview:**

This paper presents the use of Transformer models on an important application of OCRing documents into latex format. The paper will have a high utility to the scientific community and overall communities who rely on documents in image format. While one reviewer is not impressed by the work so much (and considers it is a simple application of transformer), I think the value of the experiments and the work done is high. In addition, other reviewers are more positive about the paper.

**Justification For Why Not Higher Score:**

The paper is a direct application of transformer, so in some senes, there is limited "technical novelty".

**Justification For Why Not Lower Score:**

I do think the paper should be accepted.

---

### Decision · Program_Chairs · 2024-01-16

Accept (poster)